# Recurrent Kernel Networks

**Dexiong Chen**
Inria*
dexiong.chen@inria.fr

**Laurent Jacob**
CNRS†
laurent.jacob@univ-lyon1.fr

**Julien Mairal**
Inria*
julien.mairal@inria.fr

## Abstract

Substring kernels are classical tools for representing biological sequences or text. However, when large amounts of annotated data are available, models that allow end-to-end training such as neural networks are often preferred. Links between recurrent neural networks (RNNs) and substring kernels have recently been drawn, by formally showing that RNNs with specific activation functions were points in a reproducing kernel Hilbert space (RKHS). In this paper, we revisit this link by generalizing convolutional kernel networks—originally related to a relaxation of the mismatch kernel—to model gaps in sequences. It results in a new type of recurrent neural network which can be trained end-to-end with backpropagation, or without supervision by using kernel approximation techniques. We experimentally show that our approach is well suited to biological sequences, where it outperforms existing methods for protein classification tasks.

## 1 Introduction

Learning from biological sequences is important for a variety of scientific fields such as evolution [8] or human health [16]. In order to use classical statistical models, a first step is often to map sequences to vectors of fixed size, while retaining relevant features for the considered learning task. For a long time, such features have been extracted from sequence alignment, either against a reference or between each others [3]. The resulting features are appropriate for sequences that are similar enough, but they become ill-defined when sequences are not suited to alignment. This includes important cases such as microbial genomes, distant species, or human diseases, and calls for alternative representations [7].

String kernels provide generic representations for biological sequences, most of which do not require global alignment [34]. In particular, a classical approach maps sequences to a huge-dimensional feature space by enumerating statistics about all occuring subsequences. These subsequences may be simple classical $k$-mers leading to the spectrum kernel [21], $k$-mers up to mismatches [22], or gap-allowing subsequences [24]. Other approaches involve kernels based on a generative model [17, 35], or based on local alignments between sequences [36] inspired by convolution kernels [11, 37].

The goal of kernel design is then to encode prior knowledge in the learning process. For instance, modeling gaps in biological sequences is important since it allows taking into account short insertion and deletion events, a common source of genetic variation. However, even though kernel methods are good at encoding prior knowledge, they provide fixed task-independent representations. When large amounts of data are available, approaches that optimize the data representation for the prediction task

are now often preferred. For instance, convolutional neural networks [19] are commonly used for DNA sequence modeling [1, 2, 41], and have been successful for natural language processing [18]. While convolution filters learned over images are interpreted as image patches, those learned over sequences are viewed as sequence motifs. RNNs such as long short-term memory networks (LSTMs) [14] are also commonly used in both biological [13] and natural language processing contexts [5, 26].

Motivated by the regularization mechanisms of kernel methods, which are useful when the amount of data is small and are yet imperfect in neural networks, hybrid approaches have been developed between the kernel and neural networks paradigms [6, 27, 40]. Closely related to our work, the convolutional kernel network (CKN) model originally developed for images [25] was successfully adapted to biological sequences in [4]. CKNs for sequences consist in a continuous relaxation of the mismatch kernel: while the latter represents a sequence by its content in $k$-mers up to a few discrete errors, the former considers a continuous relaxation, leading to an infinite-dimensional sequence representation. Finally, a kernel approximation relying on the Nyström method [38] projects the mapped sequences to a linear subspace of the RKHS, spanned by a finite number of motifs. When these motifs are learned end-to-end with backpropagation, learning with CKNs can also be thought of as performing feature selection in the—infinite dimensional—RKHS.

In this paper, we generalize CKNs for sequences by allowing gaps in motifs, motivated by genomics applications. The kernel map retains the convolutional structure of CKNs but the kernel approximation that we introduce can be computed using a recurrent network, which we call recurrent kernel network (RKN). This RNN arises from the dynamic programming structure used to compute efficiently the substring kernel of [24], a link already exploited by [20] to derive their sequence neural network, which was a source of inspiration for our work. Both our kernels rely on a RNN to build a representation of an input sequence by computing a string kernel between this sequence and a set of learnable filters. Yet, our model exhibits several differences with [20], who use the regular substring kernel of [24] and compose this representation with another non-linear map—by applying an activation function to the output of the RNN. By contrast, we obtain a different RKHS directly by relaxing the substring kernel to allow for inexact matching at the compared positions, and embed the Nyström approximation within the RNN. The resulting feature space can be interpreted as a continuous neighborhood around all substrings (with gaps) of the described sequence. Furthermore, our RNN provides a finite-dimensional approximation of the relaxed kernel, relying on the Nyström approximation method [38]. As a consequence, RKNs may be learned in an unsupervised manner (in such a case, the goal is to approximate the kernel map), and with supervision with backpropagation, which may be interpreted as performing feature selection in the RKHS.

**Contributions.**    In this paper, we make the following contributions:
• We generalize convolutional kernel networks for sequences [4] to allow gaps, an important option for biological data. As in [4], we observe that the kernel formulation brings practical benefits over traditional CNNs or RNNs [13] when the amount of labeled data is small or moderate.
• We provide a kernel point of view on recurrent neural networks with new unsupervised and supervised learning algorithms. The resulting feature map can be interpreted in terms of gappy motifs, and end-to-end learning amounts to performing feature selection.
• Based on [28], we propose a new way to simulate max pooling in RKHSs, thus solving a classical discrepancy between theory and practice in the literature of string kernels, where sums are often replaced by a maximum operator that does not ensure positive definiteness [36].

## 2    Background on Kernel Methods and String Kernels

Kernel methods consist in mapping data points living in a set $\mathcal{X}$ to a possibly infinite-dimensional Hilbert space $\mathcal{H}$, through a mapping function $\Phi : \mathcal{X} \to \mathcal{H}$, before learning a simple predictive model in $\mathcal{H}$ [33]. The so-called kernel trick allows to perform learning without explicitly computing this mapping, as long as the inner-product $K(\mathbf{x}, \mathbf{x}') = \langle \Phi(\mathbf{x}), \Phi(\mathbf{x}') \rangle_{\mathcal{H}}$ between two points $\mathbf{x}, \mathbf{x}'$ can be efficiently computed. Whereas kernel methods traditionally lack scalability since they require computing an $n \times n$ Gram matrix, where $n$ is the amount of training data, recent approaches based on approximations have managed to make kernel methods work at large scale in many cases [30, 38].

For sequences in $\mathcal{X} = \mathcal{A}^*$, which is the set of sequences of any possible length over an alphabet $\mathcal{A}$, the mapping $\Phi$ often enumerates subsequence content. For instance, the spectrum kernel maps sequences to a fixed-length vector $\Phi(\mathbf{x}) = (\phi_u(\mathbf{x}))_{u \in \mathcal{A}^k}$, where $\mathcal{A}^k$ is the set of $k$-mers—length-$k$

sequence of characters in $\mathcal{A}$ for some $k$ in $\mathbb{N}$, and $\phi_u(\mathbf{x})$ counts the number of occurrences of $u$ in $\mathbf{x}$ [21]. The mismatch kernel [22] operates similarly, but $\phi_u(\mathbf{x})$ counts the occurrences of $u$ up to a few mismatched letters, which is useful when $k$ is large and exact occurrences are rare.

## 2.1 Substring kernels

As [20], we consider the substring kernel introduced in [24], which allows to model the presence of gaps when trying to match a substring $u$ to a sequence $\mathbf{x}$. Modeling gaps requires introducing the following notation: $\mathcal{I}_{\mathbf{x},k}$ denotes the set of indices of sequence $\mathbf{x}$ with $k$ elements $(i_1, \ldots, i_k)$ satisfying $1 \leq i_1 < \cdots < i_k \leq |\mathbf{x}|$, where $|\mathbf{x}|$ is the length of $\mathbf{x}$. For an index set $\mathbf{i}$ in $\mathcal{I}_{\mathbf{x},k}$, we may now consider the subsequence $\mathbf{x_i} = (\mathbf{x}_{i_1}, \ldots, \mathbf{x}_{i_k})$ of $\mathbf{x}$ indexed by $\mathbf{i}$. Then, the substring kernel takes the same form as the mismatch and spectrum kernels, but $\phi_u(\mathbf{x})$ counts all—consecutive or not—subsequences of $\mathbf{x}$ equal to $u$, and weights them by the number of gaps. Formally, we consider a parameter $\lambda$ in $[0, 1]$, and $\phi_u(\mathbf{x}) = \sum_{\mathbf{i} \in \mathcal{I}_{\mathbf{x},k}} \lambda^{\mathrm{gaps}(\mathbf{i})} \delta(u, \mathbf{x_i})$, where $\delta(u, v) = 1$ if and only if $u = v$, and $0$ otherwise, and $\mathrm{gaps}(\mathbf{i}) := i_k - i_1 - k + 1$ is the number of gaps in the index set $\mathbf{i}$. When $\lambda$ is small, gaps are heavily penalized, whereas a value close to $1$ gives similar weights to all occurrences. Ultimately, the resulting kernel between two sequences $\mathbf{x}$ and $\mathbf{x}'$ is

$$\mathcal{K}^s(\mathbf{x}, \mathbf{x}') := \sum_{\mathbf{i} \in \mathcal{I}_{\mathbf{x},k}} \sum_{\mathbf{j} \in \mathcal{I}_{\mathbf{x}',k}} \lambda^{\mathrm{gaps}(\mathbf{i})} \lambda^{\mathrm{gaps}(\mathbf{j})} \delta\left(\mathbf{x_i}, \mathbf{x'_j}\right). \tag{1}$$

As we will see in Section 3, our RKN model relies on (1), but unlike [20], we replace the quantity $\delta(\mathbf{x_i}, \mathbf{x'_j})$ that matches exact occurrences by a relaxation, allowing more subtle comparisons. Then, we will show that the model can be interpreted as a gap-allowed extension of CKNs for sequences. We also note that even though $\mathcal{K}^s$ seems computationally expensive at first sight, it was shown in [24] that (1) admits a dynamic programming structure leading to efficient computations.

## 2.2 The Nyström method

When computing the Gram matrix is infeasible, it is typical to use kernel approximations [30, 38], consisting in finding a $q$-dimensional mapping $\psi : \mathcal{X} \to \mathbb{R}^q$ such that the kernel $K(\mathbf{x}, \mathbf{x}')$ can be approximated by a Euclidean inner-product $\langle \psi(\mathbf{x}), \psi(\mathbf{x}') \rangle_{\mathbb{R}^q}$. Then, kernel methods can be simulated by a linear model operating on $\psi(\mathbf{x})$, which does not raise scalability issues if $q$ is reasonably small. Among kernel approximations, the Nyström method consists in projecting points of the RKHS onto a $q$-dimensional subspace, allowing to represent points into a $q$-dimensional coordinate system.

Specifically, consider a collection of $Z = \{\mathbf{z}_1, \ldots, \mathbf{z}_q\}$ points in $\mathcal{X}$ and consider the subspace

$$\mathcal{E} = \mathrm{Span}(\Phi(\mathbf{z}_1), \ldots, \Phi(\mathbf{z}_q)) \quad \text{and define} \quad \psi(\mathbf{x}) = K_{ZZ}^{-\frac{1}{2}} K_Z(\mathbf{x}),$$

where $K_{ZZ}$ is the $q \times q$ Gram matrix of $K$ restricted to the samples $\mathbf{z}_1, \ldots, \mathbf{z}_q$ and $K_Z(\mathbf{x})$ in $\mathbb{R}^q$ carries the kernel values $K(\mathbf{x}, \mathbf{z}_j), j = 1, \ldots, q$. This approximation only requires $q$ kernel evaluations and often retains good performance for learning. Interestingly as noted in [25], $\langle \psi(\mathbf{x}), \psi(\mathbf{x}') \rangle_{\mathbb{R}^q}$ is exactly the inner-product in $\mathcal{H}$ between the projections of $\Phi(\mathbf{x})$ and $\Phi(\mathbf{x}')$ onto $\mathcal{E}$, which remain in $\mathcal{H}$.

When $\mathcal{X}$ is a Euclidean space—this can be the case for sequences when using a one-hot encoding representation, as discussed later— a good set of anchor points $\mathbf{z}_j$ can be obtained by simply clustering the data and choosing the centroids as anchor points [39]. The goal is then to obtain a subspace $\mathcal{E}$ that spans data as best as possible. Otherwise, previous works on kernel networks [4, 25] have also developed procedures to learn the set of anchor points end-to-end by optimizing over the learning objective. This approach can then be seen as performing feature selection in the RKHS.

# 3 Recurrent Kernel Networks

With the previous tools in hand, we now introduce RKNs. We show that it admits variants of CKNs, substring and local alignment kernels as special cases, and we discuss its relation with RNNs.

## 3.1 A continuous relaxation of the substring kernel allowing mismatches

From now on, and with an abuse of notation, we represent characters in $\mathcal{A}$ as vectors in $\mathbb{R}^d$. For instance, when using one-hot encoding, a DNA sequence $\mathbf{x} = (\mathbf{x}_1, \ldots, \mathbf{x}_m)$ of length $m$ can be seen

as a 4-dimensional sequence where each $\mathbf{x}_j$ in $\{0, 1\}^4$ has a unique non-zero entry indicating which of $\{A, C, G, T\}$ is present at the $j$-th position, and we denote by $\mathcal{X}$ the set of such sequences. We now define the single-layer RKN as a generalized substring kernel (1) in which the indicator function $\delta(\mathbf{x_i}, \mathbf{x'_j})$ is replaced by a kernel for $k$-mers:

$$\mathcal{K}_k(\mathbf{x}, \mathbf{x'}) := \sum_{\mathbf{i} \in \mathcal{I}_{\mathbf{x},k}} \sum_{\mathbf{j} \in \mathcal{I}_{\mathbf{x'},k}} \lambda_{\mathbf{x},\mathbf{i}} \lambda_{\mathbf{x},\mathbf{j}} e^{-\frac{\alpha}{2} \|\mathbf{x_i} - \mathbf{x'_j}\|^2}, \tag{2}$$

where we assume that the vectors representing characters have unit $\ell_2$-norm, such that $e^{-\frac{\alpha}{2}\|\mathbf{x_i}-\mathbf{x'_j}\|^2} = e^{\alpha(\langle \mathbf{x_i}, \mathbf{x'_j}\rangle - k)} = \prod_{t=1}^{k} e^{\alpha(\langle \mathbf{x}_{i_t}, \mathbf{x'}_{j_t}\rangle - 1)}$ is a dot-product kernel, and $\lambda_{\mathbf{x},\mathbf{i}} = \lambda^{\text{gaps}(\mathbf{i})}$ if we follow (1).

For $\lambda = 0$ and using the convention $0^0 = 1$, all the terms in these sums are zero except those for $k$-mers with no gap, and we recover the kernel of the CKN model of [4] with a convolutional structure—up to the normalization, which is done $k$-mer-wise in CKN instead of position-wise.

Compared to (1), the relaxed version (2) accommodates inexact $k$-mer matching. This is important for protein sequences, where it is common to consider different similarities between amino acids in terms of substitution frequency along evolution [12]. This is also reflected in the underlying sequence representation in the RKHS illustrated in Figure 1: by considering $\varphi(.)$ the kernel mapping and RKHS $\mathcal{H}$ such that $K(\mathbf{x_i}, \mathbf{x'_j}) = e^{-\frac{\alpha}{2}\|\mathbf{x_i}-\mathbf{x'_j}\|^2} = \langle \varphi(\mathbf{x_i}), \varphi(\mathbf{x'_j})\rangle_{\mathcal{H}}$, we have

$$\mathcal{K}_k(\mathbf{x}, \mathbf{x'}) = \left\langle \sum_{\mathbf{i} \in \mathcal{I}_{\mathbf{x},k}} \lambda_{\mathbf{x},\mathbf{i}} \varphi(\mathbf{x_i}), \sum_{\mathbf{j} \in \mathcal{I}_{\mathbf{x'},k}} \lambda_{\mathbf{x},\mathbf{j}} \varphi(\mathbf{x'_j}) \right\rangle_{\mathcal{H}}. \tag{3}$$

A natural feature map for a sequence $\mathbf{x}$ is therefore $\Phi_k(\mathbf{x}) = \sum_{\mathbf{i} \in \mathcal{I}_{\mathbf{x},k}} \lambda_{\mathbf{x},\mathbf{i}} \varphi(\mathbf{x_i})$: using the RKN amounts to representing $\mathbf{x}$ by a mixture of continuous neighborhoods $\varphi(\mathbf{x_i}) : \mathbf{z} \mapsto e^{-\frac{\alpha}{2}\|\mathbf{x_i}-\mathbf{z}\|^2}$ centered on all its $k$-subsequences $\mathbf{x_i}$, each weighted by the corresponding $\lambda_{\mathbf{x},\mathbf{i}}$ (e.g., $\lambda_{\mathbf{x},\mathbf{i}} = \lambda^{\text{gaps}(\mathbf{i})}$). As a particular case, a feature map of CKN [4] is the sum of the kernel mapping of all the $k$-mers without gap.

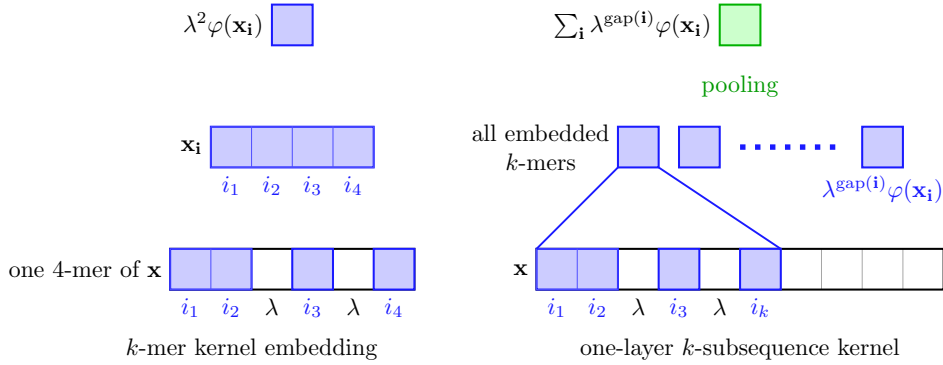

Figure 1: Representation of sequences in a RKHS based on $\mathcal{K}_k$ with $k = 4$ and $\lambda_{\mathbf{x},\mathbf{i}} = \lambda^{\text{gaps}(\mathbf{i})}$.

## 3.2 Extension to all $k$-mers and relation to the local alignment kernel

Dependency in the hyperparameter $k$ can be removed by summing $\mathcal{K}_k$ over all possible values:

$$\mathcal{K}_{\text{sum}}(\mathbf{x}, \mathbf{x'}) := \sum_{k=1}^{\infty} \mathcal{K}_k(\mathbf{x}, \mathbf{x'}) = \sum_{k=1}^{\max(|\mathbf{x}|,|\mathbf{x'}|)} \mathcal{K}_k(\mathbf{x}, \mathbf{x'}).$$

Interestingly, we note that $\mathcal{K}_{\text{sum}}$ admits the local alignment kernel of [36] as a special case. More precisely, local alignments are defined via the tensor product set $\mathcal{A}_k(\mathbf{x}, \mathbf{x'}) := \mathcal{I}_{\mathbf{x},k} \times \mathcal{I}_{\mathbf{x'},k}$, which contains all possible alignments of $k$ positions between a pair of sequences $(\mathbf{x}, \mathbf{x'})$. The local alignment score of each such alignment $\pi = (\mathbf{i}, \mathbf{j})$ in $\mathcal{A}_k(\mathbf{x}, \mathbf{x'})$ is defined, by [36], as $S(\mathbf{x}, \mathbf{x'}, \pi) := \sum_{t=1}^{k} s(\mathbf{x}_{i_t}, \mathbf{x'}_{j_t}) - \sum_{t=1}^{k-1} [g(i_{t+1} - i_t - 1) + g(j_{t+1} - j_t - 1)]$, where $s$ is a symmetric substitution

function and $g$ is a gap penalty function. The local alignment kernel in [36] can then be expressed in terms of the above local alignment scores (Thrm. 1.7 in [36]):

$$K_{LA}(\mathbf{x}, \mathbf{x}') = \sum_{k=1}^{\infty} K_{LA}^k(\mathbf{x}, \mathbf{x}') := \sum_{k=1}^{\infty} \sum_{\pi \in \mathcal{A}_k(\mathbf{x}, \mathbf{x}')} \exp(\beta \mathcal{S}(\mathbf{x}, \mathbf{x}', \pi)) \quad \text{for some } \beta > 0. \quad (4)$$

When the gap penalty function is linear—that is, $g(x) = cx$ with $c > 0$, $K_{LA}^k$ becomes $K_{LA}^k(\mathbf{x}, \mathbf{x}') = \sum_{\pi \in \mathcal{A}_k(\mathbf{x}, \mathbf{x}')} \exp(\beta \mathcal{S}(\mathbf{x}, \mathbf{x}', \pi)) = \sum_{(\mathbf{i},\mathbf{j}) \in \mathcal{A}_k(\mathbf{x}, \mathbf{x}')} e^{-c\beta \text{gaps}(\mathbf{i})} e^{-c\beta \text{gaps}(\mathbf{j})} \prod_{t=1}^k e^{\beta s(\mathbf{x}_{i_t}, \mathbf{x}'_{j_t})}$. When $s(\mathbf{x}_{i_t}, \mathbf{x}'_{j_t})$ can be written as an inner-product $\langle \psi_s(\mathbf{x}_{i_t}), \psi_s(\mathbf{x}'_{j_t}) \rangle$ between normalized vectors, we see that $K_{LA}$ becomes a special case of (2)—up to a constant factor—with $\lambda_{\mathbf{x},\mathbf{i}} = e^{-c\beta \text{gaps}(\mathbf{i})}$, $\alpha = \beta$.

This observation sheds new lights on the relation between the substring and local alignment kernels, which will inspire new algorithms in the sequel. To the best of our knowledge, the link we will provide between RNNs and local alignment kernels is also new.

### 3.3 Nyström approximation and recurrent neural networks

As in CKNs, we now use the Nyström approximation method as a building block to make the above kernels tractable. According to (3), we may first use the Nyström method described in Section 2.2 to find an approximate embedding for the quantities $\varphi(\mathbf{x}_{\mathbf{i}})$, where $\mathbf{x}_{\mathbf{i}}$ is one of the $k$-mers represented as a matrix in $\mathbb{R}^{k \times d}$. This is achieved by choosing a set $Z = \{\mathbf{z}_1, \dots, \mathbf{z}_q\}$ of anchor points in $\mathbb{R}^{k \times d}$, and by encoding $\varphi(\mathbf{x}_{\mathbf{i}})$ as $K_{ZZ}^{-1/2} K_Z(\mathbf{x}_{\mathbf{i}})$—where $K$ is the kernel of $\mathcal{H}$. Such an approximation for $k$-mers yields the $q$-dimensional embedding for the sequence $\mathbf{x}$:

$$\psi_k(\mathbf{x}) = \sum_{\mathbf{i} \in \mathcal{I}_{\mathbf{x},k}} \lambda_{\mathbf{x},\mathbf{i}} K_{ZZ}^{-\frac{1}{2}} K_Z(\mathbf{x}_{\mathbf{i}}) = K_{ZZ}^{-\frac{1}{2}} \sum_{\mathbf{i} \in \mathcal{I}_{\mathbf{x},k}} \lambda_{\mathbf{x},\mathbf{i}} K_Z(\mathbf{x}_{\mathbf{i}}). \quad (5)$$

Then, an approximate feature map $\psi_{\text{sum}}(\mathbf{x})$ for the kernel $\mathcal{K}_{\text{sum}}$ can be obtained by concatenating the embeddings $\psi_1(\mathbf{x}), \dots, \psi_k(\mathbf{x})$ for $k$ large enough.

**The anchor points as motifs.** The continuous relaxation of the substring kernel presented in (2) allows us to learn anchor points that can be interpreted as sequence motifs, where each position can encode a mixture of letters. This can lead to more relevant representations than $k$-mers for learning on biological sequences. For example, the fact that a DNA sequence is bound by a particular transcription factor can be associated with the presence of a T followed by either a G or an A, followed by another T, would require two $k$-mers but a single motif [4]. Our kernel is able to perform such a comparison.

**Efficient computations of $\mathcal{K}_k$ and $\mathcal{K}_{\text{sum}}$ approximation via RNNs.** A naive computation of $\psi_k(\mathbf{x})$ would require enumerating all substrings present in the sequence, which may be exponentially large when allowing gaps. For this reason, we use the classical dynamic programming approach of substring kernels [20, 24]. Consider then the computation of $\psi_j(\mathbf{x})$ defined in (5) for $j = 1, \dots, k$ as well as a set of anchor points $Z_k = \{\mathbf{z}_1, \dots, \mathbf{z}_q\}$ with the $\mathbf{z}_i$'s in $\mathbb{R}^{d \times k}$. We also denote by $Z_j$ the set obtained when keeping only $j$-th first positions (columns) of the $\mathbf{z}_j$'s, leading to $Z_j = \{[\mathbf{z}_1]_{1:j}, \dots, [\mathbf{z}_q]_{1:j}\}$, which will serve as anchor points for the kernel $\mathcal{K}_j$ to compute $\psi_j(\mathbf{x})$. Finally, we denote by $\mathbf{z}_i^j$ in $\mathbb{R}^d$ the $j$-th column of $\mathbf{z}_i$ such that $\mathbf{z}_i = [\mathbf{z}_i^1, \dots, \mathbf{z}_i^k]$. Then, the embeddings $\psi_1(\mathbf{x}), \dots, \psi_k(\mathbf{x})$ can be computed recursively by using the following theorem:

**Theorem 1.** *For any $j \in \{1, \dots, k\}$ and $t \in \{1, \dots, |\mathbf{x}|\}$,*

$$\psi_j(\mathbf{x}_{1:t}) = K_{Z_j Z_j}^{-\frac{1}{2}} \begin{cases} \mathbf{c}_j[t] & \text{if } \lambda_{\mathbf{x},\mathbf{i}} = \lambda^{|\mathbf{x}| - i_1 - j + 1}, \\ \mathbf{h}_j[t] & \text{if } \lambda_{\mathbf{x},\mathbf{i}} = \lambda^{gaps(\mathbf{i})}, \end{cases} \quad (6)$$

*where $c_j[t]$ and $h_j[t]$ form a sequence of vectors in $\mathbb{R}^q$ indexed by $t$ such that $c_j[0] = h_j[0] = 0$, and $c_0[t]$ is a vector that contains only ones, while the sequence obeys the recursion*

$$\begin{aligned} \mathbf{c}_j[t] &= \lambda \mathbf{c}_j[t-1] + \mathbf{c}_{j-1}[t-1] \odot \mathbf{b}_j[t] & 1 \le j \le k, \\ \mathbf{h}_j[t] &= \mathbf{h}_j[t-1] + \mathbf{c}_{j-1}[t-1] \odot \mathbf{b}_j[t] & 1 \le j \le k, \end{aligned} \quad (7)$$

*where $\odot$ is the elementwise multiplication operator and $\mathbf{b}_j[t]$ is a vector in $\mathbb{R}^q$ whose entry $i$ in $\{1, \dots, q\}$ is $e^{-\frac{\alpha}{2} \|\mathbf{x}_t - \mathbf{z}_j^i\|^2} = e^{\alpha(\langle \mathbf{x}_t, \mathbf{z}_j^i \rangle - 1)}$ and $\mathbf{x}_t$ is the $t$-th character of $\mathbf{x}$.*

A proof is provided in Appendix A and is based on classical recursions for computing the substring kernel, which were interpreted as RNNs by [20]. The main difference in the RNN structure we obtain is that their non-linearity is applied over the outcome of the network, leading to a feature map formed by composing the feature map of the substring kernel of [24] and another one from a RKHS that contains their non-linearity. By contrast, our non-linearities are built explicitly in the substring kernel, by relaxing the indicator function used to compare characters. The resulting feature map is a continuous neighborhood around all substrings of the described sequence. In addition, the Nyström method yields an orthogonalization factor $K_{ZZ}^{-1/2}$ to the output $K_Z(\mathbf{x})$ of the network to compute our approximation, which is perhaps the only non-standard component of our RNN. This factor provides an interpretation of $\psi(\mathbf{x})$ as a kernel approximation. As discussed next, it makes it possible to learn the anchor points by $k$-means, see [4], which also makes the initialization of the supervised learning procedure simple without having to deal with the scaling of the initial motifs/filters $\mathbf{z}_j$.

**Learning the anchor points** $Z$. We now turn to the application of RKNs to supervised learning. Given $n$ sequences $\mathbf{x}^1, \ldots, \mathbf{x}^n$ in $\mathcal{X}$ and their associated labels $y^1, \ldots, y^n$ in $\mathcal{Y}$, e.g., $\mathcal{Y} = \{-1, 1\}$ for binary classification or $\mathcal{Y} = \mathbb{R}$ for regression, our objective is to learn a function in the RKHS $\mathcal{H}$ of $\mathcal{K}_k$ by minimizing

$$\min_{f \in \mathcal{H}} \frac{1}{n} \sum_{i=1}^{n} L(f(\mathbf{x}^i), y^i) + \frac{\mu}{2}\|f\|_{\mathcal{H}}^2,$$

where $L : \mathbb{R} \times \mathbb{R} \to \mathbb{R}$ is a convex loss function that measures the fitness of a prediction $f(\mathbf{x}^i)$ to the true label $y^i$ and $\mu$ controls the smoothness of the predictive function. After injecting our kernel approximation $\mathcal{K}_k(\mathbf{x}, \mathbf{x}') \simeq \langle \psi_k(\mathbf{x}), \psi_k(\mathbf{x}') \rangle_{\mathbb{R}^q}$, the problem becomes

$$\min_{\mathbf{w} \in \mathbb{R}^q} \frac{1}{n} \sum_{i=1}^{n} L\left(\langle \psi_k(\mathbf{x}^i), \mathbf{w} \rangle, y^i\right) + \frac{\mu}{2}\|\mathbf{w}\|^2. \tag{8}$$

Following [4, 25], we can learn the anchor points $Z$ without exploiting training labels, by applying a $k$-means algorithm to all (or a subset of) the $k$-mers extracted from the database and using the obtained centroids as anchor points. Importantly, once $Z$ has been obtained, the linear function parametrized by $\mathbf{w}$ is still optimized with respect to the supervised objective (8). This procedure can be thought of as learning a general representation of the sequences disregarding the supervised task, which can lead to a relevant description while limiting overfitting.

Another strategy consists in optimizing (8) jointly over $(Z, \mathbf{w})$, after observing that $\psi_k(\mathbf{x}) = K_{ZZ}^{-1/2} \sum_{\mathbf{i} \in \mathcal{I}_{\mathbf{x},k}} \lambda_{\mathbf{x},\mathbf{i}} K_Z(\mathbf{x_i})$ is a smooth function of $Z$. Learning can be achieved by using backpropagation over $(Z, \mathbf{w})$, or by using an alternating minimization strategy between $Z$ and $\mathbf{w}$. It leads to an end-to-end scheme where both the representation and the function defined over this representation are learned with respect to the supervised objective (8). Backpropagation rules for most operations are classical, except for the matrix inverse square root function, which is detailed in Appendix B. Initialization is also parameter-free since the unsupervised learning approach may be used for that.

### 3.4 Extensions

**Multilayer construction.** In order to account for long-range dependencies, it is possible to construct a multilayer model based on kernel compositions similar to [20]. Assume that $\mathcal{K}_k^{(n)}$ is the $n$-th layer kernel and $\Phi_k^{(n)}$ its mapping function. The corresponding $(n+1)$-th layer kernel is defined as

$$\mathcal{K}_k^{(n+1)}(\mathbf{x}, \mathbf{x}') = \sum_{\mathbf{i} \in \mathcal{I}_{\mathbf{x},k}, \mathbf{j} \in \mathcal{I}_{\mathbf{x}',k}} \lambda_{\mathbf{x},\mathbf{i}}^{(n+1)} \lambda_{\mathbf{x}',\mathbf{j}}^{(n+1)} \prod_{t=1}^{k} K_{n+1}(\Phi_k^{(n)}(\mathbf{x}_{1:i_t}), \Phi_k^{(n)}(\mathbf{x}'_{1:j_t})), \tag{9}$$

where $K_{n+1}$ will be defined in the sequel and the choice of weights $\lambda_{\mathbf{x},\mathbf{i}}^{(n)}$ slightly differs from the single-layer model. We choose indeed $\lambda_{\mathbf{x},\mathbf{i}}^{(N)} = \lambda^{\text{gaps}(\mathbf{i})}$ only for the last layer $N$ of the kernel, which depends on the number of gaps in the index set $\mathbf{i}$ but not on the index positions. Since (9) involves a kernel $K_{n+1}$ operating on the representation of prefix sequences $\Phi_k^{(n)}(\mathbf{x}_{1:t})$ from layer $n$, the representation makes sense only if $\Phi_k^{(n)}(\mathbf{x}_{1:t})$ carries mostly local information close to position $t$.

Otherwise, information from the beginning of the sequence would be overrepresented. Ideally, we would like the range-dependency of $\Phi_k^{(n)}(\mathbf{x}_{1:t})$ (the size of the window of indices before $t$ that influences the representation, akin to receptive fields in CNNs) to grow with the number of layers in a controllable manner. This can be achieved by choosing $\lambda_{\mathbf{x},\mathbf{i}}^{(n)} = \lambda^{|\mathbf{x}|-i_1-k+1}$ for $n < N$, which assigns exponentially more weights to the $k$-mers close to the end of the sequence.

For the first layer, we recover the single-layer network $\mathcal{K}_k$ defined in (2) by defining $\Phi_k^{(0)}(\mathbf{x}_{1:i_k}) = \mathbf{x}_{i_k}$ and $K_1(\mathbf{x}_{i_k}, \mathbf{x}'_{j_k}) = e^{\alpha(\langle \mathbf{x}_{i_k}, \mathbf{x}'_{j_k}\rangle - 1)}$. For $n > 1$, it remains to define $K_{n+1}$ to be a homogeneous dot-product kernel, as used for instance in CKNs [25]:

$$K_{n+1}(\mathbf{u}, \mathbf{u}') = \|\mathbf{u}\|_{\mathcal{H}_n}\|\mathbf{u}\|_{\mathcal{H}_n}\kappa_n\left(\left\langle \frac{\mathbf{u}}{\|\mathbf{u}\|_{\mathcal{H}_n}}, \frac{\mathbf{u}'}{\|\mathbf{u}'\|_{\mathcal{H}_n}}\right\rangle_{\mathcal{H}_n}\right) \quad \text{with} \quad \kappa_n(t) = e^{\alpha_n(t-1)}. \quad (10)$$

Note that the Gaussian kernel $K_1$ used for 1st layer may also be written as (10) since characters are normalized. As for CKNs, the goal of homogenization is to prevent norms to grow/vanish exponentially fast with $n$, while dot-product kernels lend themselves well to neural network interpretations.

As detailed in Appendix C, extending the Nyström approximation scheme for the multilayer construction may be achieved in the same manner as with CKNs—that is, we learn one approximate embedding $\psi_k^{(n)}$ at each layer, allowing to replace the inner-products $\langle \Phi_k^{(n)}(\mathbf{x}_{1:i_t}), \Phi_k^{(n)}(\mathbf{x}'_{1:j_t})\rangle$ by their approximations $\langle \psi_k^{(n)}(\mathbf{x}_{1:i_t}), \psi_k^{(n)}(\mathbf{x}'_{1:j_t})\rangle$, and it is easy to show that the interpretation in terms of RNNs is still valid since $\mathcal{K}_k^{(n)}$ has the same sum structure as (2).

**Max pooling in RKHS.** Alignment scores (e.g. Smith-Waterman) in molecular biology rely on a max operation—over the scores of all possible alignments—to compute similarities between sequences. However, using max in a string kernel usually breaks positive definiteness, even though it seems to perform well in practice. To solve such an issue, sum-exponential is used as a proxy in [32], but it leads to diagonal dominance issue and makes SVM solvers unable to learn. For RKN, the sum in (3) can also be replaced by a max

$$\mathcal{K}_k^{\max}(\mathbf{x}, \mathbf{x}') = \left\langle \max_{\mathbf{i}\in\mathcal{I}_{\mathbf{x},k}} \lambda_{\mathbf{x},\mathbf{i}}\psi_k(\mathbf{x}_\mathbf{i}), \max_{\mathbf{j}\in\mathcal{I}_{\mathbf{x}',k}} \lambda_{\mathbf{x},\mathbf{j}}\psi_k(\mathbf{x}'_\mathbf{j})\right\rangle, \quad (11)$$

which empirically seems to perform well, but breaks the kernel interpretation, as in [32]. The corresponding recursion amounts to replacing all the sum in (7) by a max.

An alternative way to aggregate local features is the generalized max pooling (GMP) introduced in [28], which can be adapted to the context of RKHSs. Assuming that before pooling $\mathbf{x}$ is embedded to a set of $N$ local features $(\varphi_1, \ldots, \varphi_N) \in \mathcal{H}^N$, GMP builds a representation $\varphi^{\text{gmp}}$ whose inner-product with all the local features $\varphi_i$ is one: $\langle \varphi_i, \varphi^{\text{gmp}}\rangle_{\mathcal{H}} = 1$, for $i = 1, \ldots, N$. $\varphi^{\text{gmp}}$ coincides with the regular max when each $\varphi$ is an element of the canonical basis of a finite representation—*i.e.*, assuming that at each position, a single feature has value 1 and all others are 0.

Since GMP is defined by a set of inner-products constraints, it can be applied to our approximate kernel embeddings by solving a linear system. This is compatible with CKN but becomes intractable for RKN which pools across $|\mathcal{I}_{\mathbf{x},k}|$ positions. Instead, we heuristically apply GMP over the set $\psi_k(\mathbf{x}_{1:t})$ for all $t$ with $\lambda_{\mathbf{x},\mathbf{i}} = \lambda^{|\mathbf{x}|-i_1-k+1}$, which can be obtained from the RNN described in Theorem 1. This amounts to composing GMP with mean poolings obtained over each prefix of $\mathbf{x}$. We observe that it performs well in our experiments. More details are provided in Appendix D.

## 4 Experiments

We evaluate RKN and compare it to typical string kernels and RNN for protein fold recognition. Pytorch code is provided with the submission and additional details given in Appendix E.

### 4.1 Protein fold recognition on SCOP 1.67

Sequencing technologies provide access to gene and, indirectly, protein sequences for yet poorly studied species. In order to predict the 3D structure and function from the linear sequence of these

proteins, it is common to search for evolutionary related ones, a problem known as homology detection. When no evolutionary related protein with known structure is available, a—more difficult—alternative is to resort to protein fold recognition. We evaluate our RKN on such a task, where the objective is to predict which proteins share a 3D structure with the query [31].

Here we consider the Structural Classification Of Proteins (SCOP) version 1.67 [29]. We follow the preprocessing procedures of [10] and remove the sequences that are more than 95% similar, yielding 85 fold recognition tasks. Each positive training set is then extended with Uniref50 to make the dataset more balanced, as proposed in [13]. The resulting dataset can be downloaded from `http://www.bioinf.jku.at/software/LSTM_protein`. The number of training samples for each task is typically around 9,000 proteins, whose length varies from tens to thousands of amino-acids. In all our experiments we use logistic loss. We measure classification performances using auROC and auROC50 scores (area under the ROC curve and up to 50% false positives).

For CKN and RKN, we evaluate both one-hot encoding of amino-acids by 20-dimensional binary vectors and an alternative representation relying on the BLOSUM62 substitution matrix [12]. Specifically in the latter case, we represent each amino-acid by the centered and normalized vector of its corresponding substitution probabilities with other amino-acids. The local alignment kernel (4), which we include in our comparison, natively uses BLOSUM62.

**Hyperparameters.** We follow the training procedure of CKN presented in [4]. Specifically, for each of the 85 tasks, we hold out one quarter of the training samples as a validation set, use it to tune $\alpha$, gap penalty $\lambda$ and the regularization parameter $\mu$ in the prediction layer. These parameters are then fixed across datasets. RKN training also relies on the alternating strategy used for CKN: we use an Adam algorithm to update anchor points, and the L-BFGS algorithm to optimize the prediction layer. We train 100 epochs for each dataset: the initial learning rate for Adam is fixed to 0.05 and is halved as long as there is no decrease of the validation loss for 5 successive epochs. We fix $k$ to 10, the number of anchor points $q$ to 128 and use single layer CKN and RKN throughout the experiments.

**Implementation details for unsupervised models.** The anchor points for CKN and RKN are learned by k-means on 30,000 extracted $k$-mers from each dataset. The resulting sequence representations are standardized by removing mean and dividing by standard deviation and are used within a logistic regression classifier. $\alpha$ in Gaussian kernel and the parameter $\lambda$ are chosen based on validation loss and are fixed across the datasets. $\mu$ for regularization is chosen by a 5-fold cross validation on each dataset. As before, we fix $k$ to 10 and the number of anchor points $q$ to 1024. Note that the performance could be improved with larger $q$ as observed in [4], at a higher computational cost.

**Comparisons and results.** The results are shown in Table 1. The blosum62 version of CKN and RKN outperform all other methods. Improvement against the mismatch and LA kernels is likely caused by end-to-end trained kernel networks learning a task-specific representation in the form of a sparse set of motifs, whereas data-independent kernels lead to learning a dense function over the set of descriptors. This difference can have a regularizing effect akin to the $\ell_1$-norm in the parametric world, by reducing the dimension of the learned linear function $w$ while retaining relevant features for the prediction task. GPkernel also learns motifs, but relies on the exact presence of discrete motifs. Finally, both LSTM and [20] are based on RNNs but are outperformed by kernel networks. The latter was designed and optimized for NLP tasks and yields a $0.4$ auROC50 on this task.

RKNs outperform CKNs, albeit not by a large margin. Interestingly, as the two kernels only differ by their allowing gaps when comparing sequences, this results suggests that this aspect is not the most important for identifying common foldings in a one versus all setting: as the learned function discriminates on fold from all others, it may rely on coarser features and not exploit more subtle ones such as gappy motifs. In particular, the advantage of the LA-kernel against its mismatch counterpart is more likely caused by other differences than gap modelling, namely using a max rather than a mean pooling of $k$-mer similarities across the sequence, and a general substitution matrix rather than a Dirac function to quantify mismatches. Consistently, within kernel networks GMP systematically outperforms mean pooling, while being slightly behind max pooling.

Additional details and results, scatter plots, and pairwise tests between methods to assess the statistical significance of our conclusions are provided in Appendix E. Note that when $k = 14$, the auROC and auROC50 further increase to 0.877 and 0.636 respectively.

Table 1: Average auROC and auROC50 for SCOP fold recognition benchmark. LA-kernel uses BLOSUM62 to compare amino acids which is a little different from our encoding approach. Details about pairwise statistical tests between methods can be found in Appendix E.

| Method | pooling | one-hot | | BLOSUM62 | |
|---|---|---|---|---|---|
| | | auROC | auROC50 | auROC | auROC50 |
| GPkernel [10] | | 0.844 | 0.514 | | |
| SVM-pairwise [23] | | 0.724 | 0.359 | – | – |
| Mismatch [22] | | 0.814 | 0.467 | | |
| LA-kernel [32] | | – | – | 0.834 | 0.504 |
| LSTM [13] | | 0.830 | 0.566 | – | – |
| CKN-seq [4] | mean | 0.827 | 0.536 | 0.843 | 0.563 |
| CKN-seq [4] | max | 0.837 | 0.572 | 0.866 | 0.621 |
| CKN-seq | GMP | 0.838 | 0.561 | 0.856 | 0.608 |
| CKN-seq (unsup)[4] | mean | 0.804 | 0.493 | 0.827 | 0.548 |
| RKN ($\lambda = 0$) | mean | 0.829 | 0.542 | 0.838 | 0.563 |
| RKN | mean | 0.829 | 0.541 | 0.840 | 0.571 |
| RKN ($\lambda = 0$) | max | 0.840 | 0.575 | 0.862 | 0.618 |
| RKN | max | 0.844 | **0.587** | **0.871** | **0.629** |
| RKN ($\lambda = 0$) | GMP | 0.840 | 0.563 | 0.855 | 0.598 |
| RKN | GMP | **0.848** | 0.570 | 0.852 | 0.609 |
| RKN (unsup) | mean | 0.805 | 0.504 | 0.833 | 0.570 |

Table 2: Classification accuracy for SCOP 2.06. The complete table with error bars can be found in Appendix E.

| Method | ♯Params | Accuracy on SCOP 2.06 | | | Level-stratified accuracy (top1/top5/top10) | | |
|---|---|---|---|---|---|---|---|
| | | top 1 | top 5 | top 10 | family | superfamily | fold |
| PSI-BLAST | - | 84.53 | 86.48 | 87.34 | 82.20/84.50/85.30 | 86.90/88.40/89.30 | 18.90/35.10/35.10 |
| DeepSF | 920k | 73.00 | 90.25 | 94.51 | 75.87/91.77/95.14 | 72.23/90.08/94.70 | 51.35/67.57/72.97 |
| CKN (128 filters) | 211k | 76.30 | 92.17 | 95.27 | 83.30/94.22/96.00 | 74.03/91.83/95.34 | 43.78/67.03/77.57 |
| CKN (512 filters) | 843k | 84.11 | 94.29 | 96.36 | **90.24/95.77/97.21** | 82.33/94.20/96.35 | 45.41/69.19/79.73 |
| RKN (128 filters) | 211k | 77.82 | 92.89 | 95.51 | 76.91/93.13/95.70 | 78.56/92.98/95.53 | 60.54/83.78/**90.54** |
| RKN (512 filters) | 843k | **85.29** | **94.95** | **96.54** | 84.31/94.80/96.74 | **85.99/95.22/96.60** | **71.35/84.86**/89.73 |

## 4.2 Protein fold classification on SCOP 2.06

We further benchmark RKN in a fold classification task, following the protocols used in [15]. Specifically, the training and validation datasets are composed of 14699 and 2013 sequences from SCOP 1.75, belonging to 1195 different folds. The test set consists of 2533 sequences from SCOP 2.06, after removing the sequences with similarity greater than 40% with SCOP 1.75. The input sequence feature is represented by a vector of 45 dimensions, consisting of a 20-dimensional one-hot encoding of the sequence, a 20-dimensional position-specific scoring matrix (PSSM) representing the profile of amino acids, a 3-class secondary structure represented by a one-hot vector and a 2-class solvent accessibility. We further normalize each type of the feature vectors to have unit $\ell_2$-norm, which is done for each sequence position. More dataset details can be found in [15]. We use mean pooling for both CKN and RKN models, as it is more stable during training for multi-class classification. The other hyperparameters are chosen in the same way as previously. More details about hyperparameter search grid can be found in Appendix E.

The accuracy results are obtained by averaging 10 different runs and are shown in Table 2, stratified by prediction difficulty (family/superfamily/fold, more details can be found in [15]). By contrast to what we observed on SCOP 1.67, RKN sometimes yields a large improvement on CKN for fold classification, especially for detecting distant homologies. This suggests that accounting for gaps does help in some fold prediction tasks, at least in a multi-class context where a single function is learned for each fold.

**Acknowledgments**

We thank the anonymous reviewers for their insightful comments and suggestions. This work has been supported by the grants from ANR (FAST-BIG project ANR-17-CE23-0011-01), by the ERC grant number 714381 (SOLARIS), and ANR 3IA MIAI@Grenoble Alpes.

## Footnotes

*Univ. Grenoble Alpes, Inria, CNRS, Grenoble INP, LJK, 38000 Grenoble, France.

†Univ. Lyon, Université Lyon 1, CNRS, Laboratoire de Biométrie et Biologie Evolutive UMR 5558, 69000 Lyon, France

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
