[Supplementary Material]

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

# A Nyström Approximation for Single-Layer RKN

We detail here the Nytröm approximation presented in Section 3.3, which we recall here for a sequence $\mathbf{x}$:

$$\psi_k(\mathbf{x}) = K_{ZZ}^{-1/2} \sum_{\mathbf{i} \in \mathcal{I}_{\mathbf{x},k}} \lambda_{\mathbf{x},\mathbf{i}} K_Z(\mathbf{x_i}). \tag{12}$$

Consider then the computation of $\psi_j(\mathbf{x})$ defined in (12) for $j = 1, \dots, k$ given a set of anchor points $Z_k = \{\mathbf{z}_1, \dots, \mathbf{z}_q\}$ with the $\mathbf{z}_i$'s in $\mathbb{R}^{d \times k}$. Given the notations introduced in Section 3.3, we are now in shape to prove Theorem 1.

*Proof.* The proof is based on Theorem 1 of [20] and definition 2 of [24]. For $\mathbf{i} \in \mathcal{I}_{\mathbf{x},j}$, let us denote by $\mathbf{i}' = (i_1, \dots, i_{j-1})$ the $j-1$ first entries of $\mathbf{i}$. We first notice that for the Gaussian kernel $K$, we have the following factorization relation for $i = 1, \dots, q$

$$
\begin{aligned}
K(\mathbf{x_i}, [\mathbf{z}_i]_{1:j}) &= e^{\alpha(\langle \mathbf{x_i}, [\mathbf{z}_i]_{1:j} \rangle - j)} \\
&= e^{\alpha(\langle \mathbf{x_{i'}}, [\mathbf{z}_i]_{1:j-1} \rangle - (j-1))} e^{\alpha(\langle \mathbf{x}_{i_j}, \mathbf{z}_j \rangle - 1)} \\
&= K(\mathbf{x_{i'}}, [\mathbf{z}_i]_{1:j-1}) e^{\alpha(\langle \mathbf{x}_{i_j}, \mathbf{z}_j \rangle - 1)}.
\end{aligned}
$$

Thus

$$K_{Z_j}(\mathbf{x_i}) = K_{Z_{j-1}}(\mathbf{x_{i'}}) \odot \mathbf{b}_j[i_j],$$

with $\mathbf{b}_j[t]$ defined as in the theorem.

Let us denote $\sum_{\mathbf{i} \in \mathcal{I}_{\mathbf{x}_{1:t},j}} \lambda_{\mathbf{x}_{1:t},\mathbf{i}} K_{Z_j}(\mathbf{x_i})$ by $\tilde{\mathbf{c}}_j[t]$ if $\lambda_{\mathbf{x},\mathbf{i}} = \lambda^{|\mathbf{x}|-i_1-j+1}$ and by $\tilde{\mathbf{h}}_j[t]$ if $\lambda_{\mathbf{x},\mathbf{i}} = \lambda^{\mathrm{gaps}(\mathbf{i})}$. We want to prove that $\tilde{\mathbf{c}}_j[t] = \mathbf{c}_j[t]$ and $\tilde{\mathbf{h}}_j[t] = \mathbf{h}_j[t]$. First, it is clear that $\tilde{\mathbf{c}}_j[0] = 0$ for any $j$. We show by induction on $j$ that $\tilde{\mathbf{c}}_j[t] = \mathbf{c}_j[t]$. When $j = 1$, we have

$$
\begin{aligned}
\tilde{\mathbf{c}}_1[t] &= \sum_{1 \le i_1 \le t} \lambda^{t-i_1} K_{Z_1}(\mathbf{x}_{i_1}) \\
&= \sum_{1 \le i_1 \le t-1} \lambda^{t-i_1} K_{Z_1}(\mathbf{x}_{i_1}) + K_{Z_1}(\mathbf{x}_t), \\
&= \lambda \tilde{\mathbf{c}}_1[t-1] + \mathbf{b}_1[t].
\end{aligned}
$$

$\tilde{\mathbf{c}}_1[t]$ and $\mathbf{c}_1[t]$ have the same recursion and initial state thus are identical. When $j > 1$ and suppose that $\tilde{\mathbf{c}}_{j-1}[t] = \mathbf{c}_{j-1}[t]$, then we have

$$
\begin{aligned}
\tilde{\mathbf{c}}_j[t] &= \sum_{\mathbf{i} \in \mathcal{I}_{\mathbf{x}_{1:t},j}} \lambda^{t-i_1-j+1} K_{Z_j}(\mathbf{x_i}), \\
&= \underbrace{\sum_{\mathbf{i} \in \mathcal{I}_{\mathbf{x}_{1:t-1},j}} \lambda^{t-i_1-j+1} K_{Z_j}(\mathbf{x_i})}_{i_j < t} + \underbrace{\sum_{\mathbf{i}' \in \mathcal{I}_{\mathbf{x}_{1:t-1},j-1}} \lambda^{(t-1)-s_1-(j-1)+1} K_{Z_{j-1}}(\mathbf{x_{i'}}) \odot \mathbf{b}_j[t]}_{i_j = t}, \\
&= \lambda \tilde{\mathbf{c}}_j[t-1] + \tilde{\mathbf{c}}_{j-1}[t] \odot \mathbf{b}_j[t], \\
&= \lambda \tilde{\mathbf{c}}_j[t-1] + \mathbf{c}_{j-1}[t] \odot \mathbf{b}_j[t].
\end{aligned}
$$

$\tilde{\mathbf{c}}_j[t]$ and $\mathbf{c}_j[t]$ have the same recursion and initial state. We have thus proved that $\tilde{\mathbf{c}}_j[t] = c_j[t]$. Let us move on for proving $\tilde{\mathbf{h}}_j[t] = \mathbf{h}_j[t]$ by showing that they have the same initial state and recursion. It is straightforward that $\tilde{\mathbf{h}}_j[0] = 0$, then for $1 \le j \le k$ we have

$$
\begin{aligned}
\tilde{\mathbf{h}}_j[t] &= \sum_{\mathbf{i} \in \mathcal{I}_{\mathbf{x}_{1:t},j}} \lambda^{i_j-i_1-j+1} K_{Z_j}(\mathbf{x}_i), \\
&= \sum_{\mathbf{i} \in \mathcal{I}_{\mathbf{x}_{1:t-1},j}} \lambda^{i_j-i_1-j+1} K_{Z_j}(\mathbf{x_i}) + \sum_{\mathbf{i}' \in \mathcal{I}_{\mathbf{x}_{1:t-1},j-1}} \lambda^{(t-1)-s_1-(j-1)+1} K_{Z_{j-1}}(\mathbf{x_{i'}}) \odot \mathbf{b}_j[t] \\
&= \tilde{\mathbf{h}}_j[t-1] + \mathbf{c}_{j-1}[t] \odot \mathbf{b}_j[t].
\end{aligned}
$$

Therefore $\tilde{\mathbf{h}}_j[t] = \mathbf{h}_j[t]$. $\qquad\square$

# B  Back-propagation for Matrix Inverse Square Root

In Section 3.3, we have described an end-to-end scheme to jointly optimize $Z$ and $\mathbf{w}$. The back-propagation of $Z$ requires computing that of the matrix inverse square root operation as it is involved in the approximate feature map of $\mathbf{x}$ as shown in (12). The back-propagation formula is given by the following proposition, which is based on an errata of [25] and we include it here for completeness.

**Proposition 1.** *Given* $\mathbf{A}$ *a symmetric positive definite matrix in* $\mathbb{R}^{n\times n}$ *and the eigencomposition of* $\mathbf{A}$ *is written as* $\mathbf{A} = \mathbf{U}\boldsymbol{\Delta}\mathbf{U}^{\top}$ *where* $\mathbf{U}$ *is orthogonal and* $\boldsymbol{\Delta}$ *is diagonal with eigenvalues* $\delta_1, \ldots, \delta_n$. *Then*

$$d(\mathbf{A}^{-\frac{1}{2}}) = -\mathbf{U}(\mathbf{F} \circ (\mathbf{U}^{\top} d\mathbf{A}\mathbf{U}))\mathbf{U}^{\top}. \tag{13}$$

*Proof.* First, let us differentiate with respect to the inverse matrix $\mathbf{A}^{-1}$:

$$\mathbf{A}^{-1}\mathbf{A} = \mathcal{I} \quad \Longrightarrow \quad \mathbf{A}^{-1}d\mathbf{A} + d(\mathbf{A}^{-1})\mathbf{A} = 0 \quad \Longrightarrow \quad d(\mathbf{A}^{-1}) = -\mathbf{A}^{-1}d\mathbf{A}\mathbf{A}^{-1}.$$

Then, by applying the same (classical) trick,

$$\mathbf{A}^{-\frac{1}{2}}\mathbf{A}^{-\frac{1}{2}} = \mathbf{A}^{-1} \quad \Longrightarrow \quad d(\mathbf{A}^{-\frac{1}{2}})\mathbf{A}^{-\frac{1}{2}} + \mathbf{A}^{-\frac{1}{2}}d(\mathbf{A}^{-\frac{1}{2}}) = d(\mathbf{A}^{-1}) = -\mathbf{A}^{-1}d\mathbf{A}\mathbf{A}^{-1}.$$

By multiplying the last relation by $\mathbf{U}^{\top}$ on the left and by $\mathbf{U}$ on the right.

$$\mathbf{U}^{\top}d(\mathbf{A}^{-\frac{1}{2}})\mathbf{U}\boldsymbol{\Delta}^{-\frac{1}{2}} + \boldsymbol{\Delta}^{-\frac{1}{2}}\mathbf{U}^{\top}d(\mathbf{A}^{-\frac{1}{2}})\mathbf{U} = -\boldsymbol{\Delta}^{-1}\mathbf{U}^{\top}d\mathbf{A}\mathbf{U}\boldsymbol{\Delta}^{-1}.$$

Note that $\boldsymbol{\Delta}$ is diagonal. By introducing the matrix $\mathbf{F}$ such that $\mathbf{F}_{kl} = \frac{1}{\sqrt{\delta_k}\sqrt{\delta_l}(\sqrt{\delta_k}+\sqrt{\delta_l})}$, it is then easy to show that

$$\mathbf{U}^{\top}d(\mathbf{A}^{-\frac{1}{2}})\mathbf{U} = -\mathbf{F} \circ (\mathbf{U}^{\top}d\mathbf{A}\mathbf{U}),$$

where $\circ$ is the Hadamard product between matrices. Then, we are left with

$$d(\mathbf{A}^{-\frac{1}{2}}) = -\mathbf{U}(\mathbf{F} \circ (\mathbf{U}^{\top}d\mathbf{A}\mathbf{U}))\mathbf{U}^{\top}.$$

$\square$

When doing back-propagation, one is usually interested in computing a quantity $\bar{\mathbf{A}}$ such that given $\bar{\mathbf{B}}$ (with appropriate dimensions), we have

$$\langle \bar{\mathbf{B}}, d(\mathbf{A}^{-\frac{1}{2}})\rangle_F = \langle \bar{\mathbf{A}}, d\mathbf{A}\rangle_F,$$

see [9], for instance. Here, $\langle , \rangle_F$ denotes the Frobenius inner product. Then, it is easy to show that

$$\bar{\mathbf{A}} = -\mathbf{U}(\mathbf{F} \circ (\mathbf{U}^{\top}\bar{\mathbf{B}}\mathbf{U}))\mathbf{U}^{\top}.$$

# C  Multilayer Construction of RKN

For multilayer RKN, assume that we have defined $\mathcal{K}^{(n)}$ the $n$-th layer kernel. To simplify the notation below, we consider that an input sequence $\mathbf{x}$ is encoded at layer $n$ as $\mathbf{x}^{(n)} := (\Phi_k^{(n)}(\mathbf{x}_1), \Phi_k^{(n)}(\mathbf{x}_{1:2}), \ldots, \Phi_k^{(n)}(\mathbf{x}))$ where the feature map at position $t$ is $\mathbf{x}_t^{(n)} = \Phi_k^{(n)}(\mathbf{x}_{1:t})$. The $(n+1)$-layer kernel is defined by induction by

$$\mathcal{K}_k^{(n+1)}(\mathbf{x}, \mathbf{x}') = \sum_{\mathbf{i}\in\mathcal{I}_{\mathbf{x},k}, \mathbf{j}\in\mathcal{I}_{\mathbf{x}',k}} \lambda_{\mathbf{x},\mathbf{i}}^{(n)}\lambda_{\mathbf{x}',\mathbf{j}}^{(n)} \prod_{t=1}^{k} K_{n+1}(\mathbf{x}_{i_t}^{(n)}, \mathbf{x}_{j_t}'^{(n)}), \tag{14}$$

where $K_{n+1}$ is defined in (10. With the choice of weights described in Section 3.4, the construction scheme for an $n$-layer RKN is illustrated in Figure 2 The Nyström approximation scheme for multilayer RKN is straightforward by inductively applying the Nytröm method to the kernels $\mathcal{K}^{(1)}, \ldots, \mathcal{K}^{(n)}$ from bottom to top layers. Specifically, assume that $\mathcal{K}^{(n)}(\mathbf{x}, \mathbf{x}')$ is approximated by $\langle \psi_k^{(n)}(\mathbf{x}), \psi_k^{(n)}(\mathbf{x}')\rangle_{\mathbb{R}^{q_n}}$ such that the approximate feature map of $\mathbf{x}^{(n)}$ at position $t$ is $\psi_k^{(n)}(\mathbf{x}_{1:t})$. Now Consider a set of anchor points $Z_k = \{\mathbf{z}_1, \ldots, \mathbf{z}_{q_{n+1}}\}$ with the $\mathbf{z}_i$'s in $\mathbb{R}^{q_n\times k}$ which have unit norm at each column. We use the same notations as in single-layer construction. Then very similar to the single-layer RKN, the embeddings $(\psi_j^{(n+1)}(\mathbf{x}_{1:t}^{(n)}))_{1=1,\ldots,k,t=1,\ldots,|\mathbf{x}^{(n)}|}$ are given by the following recursion

Figure 2: Multilayer construction of RKN: an example with $k = 4$.

**Theorem 2.** *For any $j \in \{1, \dots, k\}$ and $t \in \{1, \dots, |\mathbf{x}^{(n)}|\}$,*

$$\psi_j^{(n+1)}(\mathbf{x}_{1:t}^{(n)}) = K_{Z_j Z_j}^{-1/2} \begin{cases} \mathbf{c}_j[t] & \text{if } \lambda_{\mathbf{x},\mathbf{i}}^{(n)} = \lambda^{|\mathbf{x}^{(n)}| - i_1 - j + 1}, \\ \mathbf{h}_j[t] & \text{if } \lambda_{\mathbf{x},\mathbf{i}}^{(n)} = \lambda^{gaps(\mathbf{i})}, \end{cases}$$

*where $c_j[t]$ and $h_j[t]$ form a sequence of vectors in $\mathbb{R}^{q_{n+1}}$ indexed by $t$ such that $c_j[0] = h_j[0] = 0$, and $c_0[t]$ is a vector that contains only ones, while the sequence obeys the recursion*

$$\begin{aligned} \mathbf{c}_j[t] &= \lambda \mathbf{c}_j[t-1] + \mathbf{c}_{j-1}[t-1] \odot \mathbf{b}_j[t] & 1 \le j \le k, \\ \mathbf{h}_j[t] &= \mathbf{h}_j[t-1] + \mathbf{c}_{j-1}[t-1] \odot \mathbf{b}_j[t] & 1 \le j \le k, \end{aligned} \tag{15}$$

*where $\odot$ is the elementwise multiplication operator and $\mathbf{b}_j[t]$ whose entry $i$ is $K_{n+1}(\mathbf{z}_j^i, \mathbf{x}_t^{(n)}) = \|\mathbf{x}_t^{(n)}\| \kappa_n \left( \left\langle \mathbf{z}_j^i, \frac{\mathbf{x}_t^{(n)}}{\|\mathbf{x}_t^{(n)}\|} \right\rangle \right)$.*

*Proof.* The proof can be obtained by that of Theorem 1 by replacing the Gaussian kernel $e^{\alpha(\langle \mathbf{x}_t, \mathbf{z}_j^i \rangle)}$ with the kernel $K_{n+1}(\mathbf{x}_t^{(n)}, \mathbf{z}_j^i)$. $\square$

## D  Generalized Max Pooling for RKN

Assume that a sequence $\mathbf{x}$ is embedded to $(\varphi_1, \dots, \varphi_n) \in \mathcal{H}^n$ local features, as in Section 3.4. Generalized max pooling (GMP) looks for a representation $\varphi^{\text{gmp}}$ such that the inner product between this vector and all the local representations is one: $\langle \varphi_i, \varphi^{\text{gmp}} \rangle_{\mathcal{H}} = 1$, for $i = 1, \dots, n$. Assuming that each $\varphi_i$ is now represented by a vector $\psi_i$ in $\mathbb{R}^q$, the above problem can be approximately solved by search an embedding vector $\psi^{\text{gmp}}$ in $\mathbb{R}^q$ such that $\langle \psi_i, \psi^{\text{gmp}} \rangle = 1$ for $i = 1, \dots, n$. In practice, and to prevent ill-conditioned problems, as shown in [28], it is possible to solve a ridge regression problem:

$$\psi^{\text{gmp}} = \underset{\psi \in \mathbb{R}^q}{\arg \min} \|\Psi^\top \psi - \mathbf{1}\|^2 + \gamma \|\psi\|^2, \tag{16}$$

where $\Psi = [\psi_1, \dots, \psi_n] \in \mathbb{R}^{q \times n}$ and $\mathbf{1}$ denotes the $n$-dimensional vectors with only 1 as entries. The solution is simply given by $\psi^{\text{gmp}} = (\Psi \Psi^\top + \gamma I)^{-1} \Psi \mathbf{1}$. It requires inverting a $q \times q$ matrix which is usually tractable when the number of anchor points is small. In particular, when $\psi_i = K_{ZZ}^{-1/2} K_Z(\mathbf{x}_i)$ the Nyström approximation of a local feature map, we have $\Psi = K_{ZZ}^{-1/2} K_{ZX}$ with $[K_{ZX}]_{ji} = K(\mathbf{z}_j, \mathbf{x}_i)$ and thus

$$\psi^{\text{gmp}} = K_{ZZ}^{\frac{1}{2}} (K_{ZX} K_{ZX}^\top + \gamma K_{ZZ})^{-1} K_{ZX} \mathbf{1}.$$

## E  Additional Experimental Material

In this section, we provide additional details about experiments and scatter plots with pairwise statistical tests.

Figure 3: Boxplots when varying filter number $q$ (left) and filter size (right).

## E.1 Protein fold recognition on SCOP 1.67

**Hyperparameter search grids.** Here, we first provide the grids used for hyperparameter search. In our experiments, we use $\sigma$ instead of $\alpha$ such that $\alpha = 1/k\sigma^2$. The search range is specified in Table 3.

Table 3: Hyperparameter search range.

| hyperparameter | search range |
|---|---|
| $\sigma$ ($\alpha = 1/k\sigma^2$) | [0.3;0.4;0.5;0.6] |
| $\mu$ for mean pooling | [1e-06;1e-05;1e-04] |
| $\mu$ for max pooling | [0.001;0.01;0.1;1.0] |
| $\lambda$ | integer multipliers of 0.05 in [0;1] |

**Comparison of unsupervised CKNs and RKNs.** Then, we provide an additional table of results to compare the unsupervised models of CKN and RKN. In this unsupervised regime, mean pooling perform better than max pooling, which is different than what we have observed in the supervised case. RKN tend to work better than CKN, while RKN-sum—that is, using the kernel $\mathcal{K}_{\mathrm{sum}}$ instead of $\mathcal{K}_k$, works better than RKN.

Table 4: Comparison of unsupervised CKN and RKN with 1024 anchor points.

| Method | Pooling | one-hot | | BLOSUM62 | |
|---|---|---|---|---|---|
| | | auROC | auROC50 | auROC | auROC50 |
| CKN | mean | 0.804 | 0.493 | 0.827 | 0.548 |
| CKN | max | 0.795 | 0.480 | 0.821 | 0.545 |
| RKN ($\lambda = 0$) | mean | 0.804 | 0.500 | 0.833 | 0.565 |
| RKN | mean | 0.805 | 0.504 | 0.833 | 0.570 |
| RKN ($\lambda = 0$) | max | 0.795 | 0.482 | 0.824 | 0.537 |
| RKN | max | 0.801 | 0.492 | 0.824 | 0.542 |
| RKN-sum ($\lambda = 0$) | mean | 0.820 | 0.526 | 0.834 | 0.567 |
| RKN-sum | mean | 0.821 | 0.527 | 0.834 | 0.565 |
| RKN-sum ($\lambda = 0$) | max | 0.825 | 0.526 | 0.837 | 0.563 |
| RKN-sum | max | 0.825 | 0.528 | 0.837 | 0.564 |

**Study of filter number $q$ and size $k$.** Here we use max pooling and fix $\sigma$ to 0.4 and $\lambda$ to 0.1. When $q$ varies $k$ is fixed to 10 and $q$ is fixed to 128 when $k$ varies. We show here the performance of RKN with different choices of $q$ and $k$. The gap hyperparameter $\lambda$ is chosen optimally for each $q$ and $k$. The results are shown in Figure 3.

**Discussion about complexity.** Performing backpropgation with our RKN model has the same complexity has a performing a similar step within a recurrent neural network, up to the computation of the inverse square root matrix $K_{ZZ}^{-1/2}$, which has complexity $O(q^3)$. When $q$ is reasonably small

$q = 128$ in our experiments, such a complexity is negligible. For instance, one forward pass with a minibatch of $b = 128$ sequences of length $m$ yields a complexity $O(k^2 mbq)$, which can typically be much greater than $q^3$.

**Computing infrastructures.** Experiments were conduced by using a shared GPU cluster, in large parts build with Nvidia gamer cards (Titan X, GTX1080TI). About 10 of these GPUs were used simultaneously to perform the experiments of this paper.

**Scatter plots and statistical testing.** Even though each method was run only one time for each task, the 85 tasks allow us to conduct statistical testing when comparing two methods. In Figures 4 and 5, we provide pairwise comparisons allowing us to assess the statistical significance of various conclusions drawn in the paper. We use a Wilcoxon signed-rank test to provide p-values.

## E.2 Protein fold classification on SCOP 2.06

**Hyperparameter search grids.** We provide the grids used for hyperparameter search, shown in Table 5.

Table 5: Hyperparameter search range for SCOP 2.06.

| hyperparameter | search range |
|---|---|
| $\sigma$ ($\alpha = 1/k\sigma^2$) | [0.3;0.4;0.5;0.6] |
| $\mu$ | [0.01;0.03;0.1;0.3;1.0;3.0;10.0] |
| $\lambda$ | integer multipliers of 0.05 in [0;1] |

**Complete results with error bars.** The classification accuracy for CKNs and RKNs on protein fold classification on SCOP 2.06 are obtained by averaging on 10 runs with different seeds. The results are shown in Table 6 with error bars.

Table 6: Classification accuracy for SCOP 2.06 on all (top) and level-stratified (bottom) test data. For CKNs and RKNs, the results are obtained over 10 different runs.

| Method | Params | Accuracy on SCOP 2.06 | | |
|---|---|---|---|---|
| | | top 1 | top 5 | top 10 |
| PSI-BLAST | - | 84.53 | 86.48 | 87.34 |
| DeepSF | 920k | 73.00 | 90.25 | 94.51 |
| CKN (128 filters) | 211k | 76.30±0.70 | 92.17±0.16 | 95.27±0.17 |
| CKN (512 filters) | 843k | 84.11±0.16 | 94.29±0.20 | 96.36±0.13 |
| RKN (128 filters) | 211k | 77.82±0.35 | 92.89±0.19 | 95.51±0.20 |
| RKN (512 filters) | 843k | 85.29±0.27 | 94.95±0.15 | 96.54±0.12 |

| Method | Level-stratified accuracy (top1/top5/top10) | | |
|---|---|---|---|
| | family | superfamily | fold |
| PSI-BLAST | 82.20/84.50/85.30 | 86.90/88.40/89.30 | 18.90/35.10/35.10 |
| DeepSF | 75.87/91.77/95.14 | 72.23/90.08/94.70 | 51.35/67.57/72.97 |
| CKN (128 filters) | 83.30±0.78/94.22±0.25/96.00±0.26 | 74.03±0.87/91.83±0.24/95.34±0.20 | 43.78±3.59/67.03±3.38/77.57±3.64 |
| CKN (512 filters) | 90.24±0.16/95.77±0.21/97.21±0.15 | 82.33±0.19/94.20±0.21/96.35±0.13 | 45.41±1.62/69.19±1.79/79.73±3.68 |
| RKN (128 filters) | 76.91±0.87/93.13±0.17/95.70±0.37 | 78.56±0.40/92.98±0.22/95.53±0.18 | 60.54±2.76/83.78±2.96/90.54±1.35 |
| RKN (512 filters) | 84.31±0.61/94.80±0.21/96.74±0.29 | 85.99±0.30/95.22±0.16/96.60±0.12 | 71.35±1.32/84.86±2.16/89.73±1.08 |

Figure 4: Scatterplots when comparing pairs of methods. In particular, we want to compare RKN vs CKN (top); , RKN vs RKN (unsup) (middle); RKN vs. LSTM (bottom).

Figure 5: Scatterplots when comparing pairs of methods. In particular, we want to compare RKN-gmp vs RKN-max (top); RKN-max vs. RKN-mean (bottom).