[Reviews · NeurIPS 2019]

Reviewer 1



The paper is well-written and clear, the different components of the RKN are introduced in an incremental fashion providing motivations for their introduction. The main concern with the paper is a lack of originality and novelty. The RKN are based on the work in [19], which presents a much more general framework for deriving neural architectures for various types of sequences and graph kernels. This work adapts the neural encoding for sequence kernels by replacing the exact match between subsequences with a gaussian kernel accounting for vectorial representations of symbols (substitution matrices for aminoacids). Note that the original work itself already mentions the possibility to replace exact matches with dot products. The multilayer construction is also taken from [19]. The authors introduce a number of efficiency refinements taken from the kernel literature, such as Nystrom approximation, mostly following another publication (ref [4]) which addresses the very problem presenting in this work. Overall, I believe the differences with respect to [19] and [4] are limited and too application specific to justify a publication in a top machine learning venue. The most recent competitor presented in the experimental evaluation, apart from the work in [4], is from 2007. Is it possible that [4] is the only recent approach dealing with fold recognition? AFTER REBUTTAL: The authors did better clarify the novelty of their contribution with respect to the papers they build upon. I still believe the scope is a bit limited, but I am not against the paper being accepted provided the authors clarify these aspects in the revised version.

Reviewer 2



In the paper titled "Recurrent Kernel Networks", the authors proposed a new method to model the gaps in comparative sequence analysis. The new method is termed Recurrent Kernel Networks, which essentially can embed sequences as points in RKHS, and leverage the training strategy of RNN. Overall, this is an innovative paper. The paper is also well organized. My only question is that in the original CKN paper, CKN was used to train DNA sequences (similar to DeepSEA), why the authors in this paper did not apply RKN to the same task?

Reviewer 3



The manuscript “Recurrent Kernel Networks” generalizes convolutional kernel networks to model gaps in sequences. It further introduces a new point of view on RNNs and proposes a new way to simulate max pooling in RKHS. In general, the manuscript is well written and introduces a nice addition to the existing literature. It shows an interesting connection between kernel methods and RNNs, which could become quite significant. At some passages it can be seen that the page limit is reducing readability, since the authors often refer to the literature instead of giving the main ideas of those papers ([4], [19], [23]). Including this information would help the readers to get a better picture of what was known before and how novel the contributions are. It is a little bit difficult to evaluate the performance improvement of the new method, since e.g., for one-hot encoding the new methods do not produce a better auROC than a standard approach (GPkernel). Furthermore, in all experiments, only one layer was used for CKNs and RKNs (line 296, page 8), which implies that the data set choice might not have been ideal for showing the performance improvement opportunities on large data sets. Furthermore, the data set is quite old (SCOP 1.75 came out in 2009 and the data is based on SCOP 1.67). Update: The author response showed additional evaluations, on newer/larger data, which substantiate the claim that the newly proposed method can also lead to performance improvements on relevant prediction problems (note: the GPkernel performances are missing in those evaluations and should be included in a final version if possible). For the RKN, the generalized max pooling is implemented by a heuristic, since it would otherwise become intractable (line 261, page 7). It performed worse than the max pooling and the performance was just shown for \lambda = 0. It would be interesting to see the performance also for other \lambdas. Update: Some new evaluations were added in the authors' response and promised for the updated appendix, solidifying the evaluations. Minor points Page 2: -line 84: the standard spectrum kernel measures the counts of the occurrences of the k-mers not just whether a k-mer is present as stated in the manuscript. -line 85 the standard mismatch kernel measures the counts of the occurrences of u up to a few letters and not just presence/absence as stated in the manuscript. Page 3: -lines 96 and 132: the gaps function might be confusing to some readers. It does not show k explicitly, but then uses it to calculate gap costs. One should make the k explicit in the call. It would also help to not introduce two ways to write the gap penalty function (lines 96, 132). -line 101: The authors spend time to introduce string kernels, but they do not put emphasis on introducing the more recent work. It would for example help the reader to see the definition of a CKN. Page 4: -line 133: It is unclear where the 0^0 comes into play and also it would help to elaborate a little bit more on the statement of line 134. Page 6: -line 223: Again, if you remove some parts at the beginning, you could mention some more details here to show how similar the construction is. Page 8: -line 295: It would be good to see stability of the performance for different choices of the parameters that were set (k, q, number of layers).

[Author Response · NeurIPS 2019]

We thank the reviewers for their insightful comments and respond to their concerns and questions below.

**Introductory remarks.** Our goal was to extend CKNs with motifs gaps. For that, we had three sources of inspiration:
- [4] is the original CKN model for sequences, which introduces links between string kernels and CNNs.
- [19] derives RNNs from the recurrent structure of substring kernels, and show that these RNNs live in a RKHS.
- [35] introduces the local alignment kernel, which appears as a special case of the kernel we define.

**Differences with respect to [19] and [4] (R1).** RKNs and [19] are undeniably related formulations, as they both exploit the recurrent structure of substring kernels to build a sequence network. Yet, as discussed below, RKNs are grounded on different principles and operate in another functional space, which we should have better explained.

RKNs and RNNs from [19] perform different computations: In [19], a non-linear function is applied on the output of recursive equations, whereas the RKN model involves a non-linear function within the recursion, see the definition of $\mathbf{b}_j$ in (7), and also involves an orthogonalization factor in (6). These constructions lead to different functional spaces and are motivated by different principles: our function space is a subspace of a RKHS defined by (2) and the Nyström approximation, while a function constructed by [19] is an element in some other RKHS. As consequences:

- RKN admits an interpretable feature map, as a weighted mixture of distance functions centered at all $k$-subsequences of the sequence (Section 3.1). No such interpretation holds for [19] (see Proposition 1 in [19]).

- The orthogonalization factor makes RKNs insensitive to the scale of the starting point, and the interpretation of RKNs filters as anchor points in the Nyström method naturally leads to an efficient unsupervised learning scheme.

- Due to the fact that RKNs provide an approximate embedding for sub-sequences, we were also able to simulate max pooling in RKHSs, bridging an old gap between theory and practice in the literature of string kernels.

Finally, RKNs are related to [4] by their relying on the relaxation of a string kernel and on the Nyström approximation. While [4] starts from the mismatch kernel, RKNs relax the subtring kernel thereby extending [4] to model gaps.

**Choice of the SCOP 1.67 benchmark and additional baselines and results (R1,R3).** Our motivation for choosing SCOP 1.67 is its extensive use as a benchmark in the kernel literature. As a consequence, most baselines reported in Table 1 are indeed a bit old. Yet, [4] presents also more recent baselines based on CNNs (with models akin to DeepBind) that perform slightly worse than CKN-seq, showing that our approach is competitive.

Following the reviewers' comments, we have benchmarked RKNs on a more recent fold recognition dataset using the protocol of "Jie Hou et al. DeepSF: deep convolutional neural network for mapping protein sequences to folds. 2018", based on SCOP 1.75 and SCOP 2.06 (see this paper for the exact description of task, dataset, and data representation). The accuracy results are shown in the next table, stratified by prediction difficulty (family/superfamily/fold). A striking conclusion is that modeling gaps (as done by RKN) is very important for the fold prediction task—closer homologies are indeed less likely to involve gaps.

| Method | ♯Params | Accuracy on SCOP2.06 | | | Level-stratified accuracy (top1/top5/top10) | | |
| --- | --- | --- | --- | --- | --- | --- | --- |
| | | top 1 | top 5 | top 10 | family | superfamily | fold |
| PSI-BLAST | - | 84.53 | 86.48 | 87.34 | 82.20/84.50/85.30 | 86.90/88.40/89.30 | 18.90/35.10/35.10 |
| DeepSF | 920k | 73.00 | 90.25 | 94.51 | 75.87/91.77/95.14 | 72.23/90.08/94.70 | 51.35/67.57/72.97 |
| CKN (128 filters) | 211k | 76.30 | 92.17 | 95.27 | 83.30/94.22/96.00 | 74.03/91.83/95.34 | 43.78/67.03/77.57 |
| CKN (512 filters) | 843k | 84.11 | 94.29 | 96.36 | 90.24/95.77/97.21 | 82.33/94.20/96.35 | 45.41/69.19/79.73 |
| RKN (128 filters) | 211k | 77.82 | 92.89 | 95.51 | 76.91/93.13/95.70 | 78.56/92.98/95.53 | 60.54/83.78/90.54 |
| RKN (512 filters) | 843k | 85.29 | 94.95 | 96.54 | 84.31/94.80/96.74 | 85.99/95.22/96.60 | 71.35/84.86/89.73 |

**Application to other DNA classification tasks (R2).** We chose not to include the DNA experiment from [4] since allowing gaps is unlikely to help predict transcription factor binding sites. Running the RKN model on this dataset shows that it performs similarly to CKN (auROC 0.805 for RKN vs 0.806 for CKN on motif occupancy datasets).

**Performance (R3).** GPkernel indeed achieves the same auROC as RKN but its auROC50 is much lower. In addition, RKNs is highly scalable, whereas GPkernel suffers from the traditional lack of scalability of kernel methods.

**Additional results (R3).** The parameters $k$ and $q$ have been chosen as in CKN-seq for fair comparison, but we agree that results for different values would be informative and will include them in the appendix. Regarding the number of layers, we have not encountered a case where a two-layer model performed better than a single-layer model involving the same number of parameters—a similar observation was made in [4]. We have also completed the experiments for GMP with $\lambda \neq 0$. Its auROC/auROC50 for one-hot is (0.848/0.57) and for BLOSUM62 (0.852/0.609).

**Interpretability and Minor points (R3).** RKN allows to capture gapped motifs. A simple way to observe these gaps would be to align the obtained motifs against a given sequence. We thank the reviewer for these remarks.

[Meta-Review · NeurIPS 2019]

This paper generalizes convolutional kernel networks to model gaps in sequences. It further introduces a new point of view on RNNs and proposes a new way to simulate max pooling in RKHS. The author rebuttal was useful to clarify the originality of the work compared to existing models, and the additional experiments shown in the rebuttal (which should be included in the final version if the paper is accepted) were important to support the relevance of the model. There were still concerns that the scope of the work is somewhat limited to biological sequence processing, but there is a healthy tradition at NeurIPS to accept application papers with novel machine learning ideas. Overall, we therefore consider this paper as a solid piece of work bringing new ML ideas and promising results on biological sequence classification.